# A preoperative predictive study of advantages of airway changes after maxillomandibular advancement surgery using computational fluid dynamics analysis

Kanako Yamagata[1,2], Keiji Shinozuka[1]*, Shouhei Ogisawa[1], Akio Himejima[3], Hiroaki Azaki[1], Shuichi Nishikubo[1], Takako Sato[1], Masaaki Suzuki[4], Tadashi Tanuma[2], Morio Tonogi[1]

1 Department of Oral and Maxillofacial Surgery, Nihon University School of Dentistry, Tokyo, Japan, 2 Laboratory of Fluid-Structural Simulation and Design, Strategic Innovation and Research Center, Teikyo University, Tokyo, Japan, 3 First Department of Oral and Maxillofacial Surgery, Osaka Dental University, Osaka, Japan, 4 Department of Otolaryngology, Teikyo University Chiba Medical Center, Chiba, Japan

* shinozuka.keiji@nihon-u.ac.jp

**Data Availability Statement:** All relevant data are within the manuscript and its Supporting Information files.

## Abstract

The purpose of this study was to develop a simulation approach for predicting maxillomandibular advancement-induced airway changes using computational fluid dynamics. Eight patients with jaw deformities who underwent maxillomandibular advancement and genioglossus advancement surgery were included in this study. Computed tomography scans and rhinomanometric readings were performed both preoperatively and postoperatively. Computational fluid dynamics models were created, and airflow simulations were performed using computational fluid dynamics software; the preferable number of computational mesh points was at least 10 million cells. The results for the right and left nares, including simulation and postoperative measurements, were qualitatively consistent, and surgery reduced airflow pressure loss. Geometry prediction simulation results were qualitatively consistent with the postoperative stereolithography data and postoperative simulation results. Simulations were performed with either the right or left naris blocked, and the predicted values were similar to those found clinically. In addition, geometry prediction simulation results were qualitatively consistent with the postoperative stereolithography data and postoperative simulation results. These findings suggest that geometry prediction simulation facilitates the preoperative prediction of the postoperative structural outcome.

## Introduction

Obstructive sleep apnea (OSA) is a highly prevalent disorder that affects 4% of the global population; it induces a frequent partial or total obstruction of the upper airway during sleep, thereby decreasing oxygen saturation and disrupting sleep [1]. In patients with diabetes and cardiovascular disorders, untreated OSA is associated with severe complications, including

**Funding:** This study was supported by the Japan Society for the Promotion of Science (KAKENHI, no. 19K10294) grants from the Dental Research Center (2019, 2020 and 2021) and Sato Fund (2018), Nihon University School of Dentistry (Tokyo, Japan).

**Competing interests:** The authors have declared that no competing interests exist.

hypertension, stroke, arrhythmia, and lower olfactory function [2–4]. As one of correlations, a mechanism which interacts with the immune system triggering pro-inflammatory pathways that represented by chronic intermittent hypoxia and sleep fragmentation has been reported in OSA patients [5]. In 1984, Riley et al. presented maxillomandibular advancement (MMA) as a new effective treatment for OSA [6]. MMA is one of the many surgical procedures that are currently available for treating OSA [7]. MMA was part of a phase 2 algorithm where it followed phase 1 (nasal, palate, tongue base procedures, hyoid advancement) [8, 9]. If phase 1 incompletely treated, then phase 2 was appropriate [10, 11]. Generally, total airway surgery for patients who include failure of other therapy or dentofacial deformity, constituting MMA with or without genioglossus advancement (GA), has shown significant success in treating patients with OSA [10–12]. In addition, Le Fort type I osteotomy and sagittal splitting ramus osteotomy are performed with MMA to anteriorly move the maxilla and mandible, and these procedures have been demonstrated to be effective [13]. MMA expands the upper airway; however, the degree of advancement required for therapeutic effectiveness is uncertain.

Standard methods for evaluating jaw movement include cephalometric analysis to determine both bone and tooth geometry in paper and mock surgery. These techniques are used to examine occlusal and facial morphological disharmony as indices for evaluation and correction. However, given that MMA is a sleep surgery, these approaches cannot predict surgical outcomes because of inability to analyze physiological functions, such as postoperative changes in upper airway geometry and respiratory volume. In addition, several uncertainties, such as the degree of advancement required to sufficiently expand the airway and reduce upper airway resistance to respiratory flow, exist when planning an MMA procedure. Airflow simulations using computational fluid dynamics (CFD) have recently been applied in OSA patients who were treated with either oral appliances [14] or adenotonsillectomy [15], maxillomandibular advancement [16], or genioglossal advancement [17]. CFD analysis may help to clarify the pathogenesis of OSA [16] and can be combined with the geometry of the pharyngeal airway, both before and after treatment, to calculate reductions in pressure and flow resistance [18]. Moreover, our previous study evaluated MMA-associated airway changes using CFD and rhinomanometry to simulate fluid dynamics in inhalation, and we found that simulated projections were comparable to actual measurements [19]. In this study, we aimed to develop a simulation approach for predicting MMA-induced airway changes using CFD.

## Materials and methods

### Participants

This study was approved by the Nihon University School of Dentistry Ethics Committee (Tokyo, Japan; approval no. EP16D007) and written informed consent was obtained from all participants. Eight patients (three men and five women; mean age, 34±8 years; range, 16–45 years) with jaw deformities who underwent MMA+GA surgery were included in this study. All operations were performed by one surgeon. Semi-rigid fixation during MMA+GA was achieved with titanium mini plates and screws. Patients underwent both rhinomanometry and three-dimensional computed tomography (3D-CT) evaluation before MMA surgery and at least 1 year after surgery. All patients provided informed consent to participate in this study. Informed consent was obtained from parents/guardians of subjects < 20 years of ages.

### CFD analyses and clinical measurement

CT scans of the nose, sinuses, and upper airway of participants were obtained with 1-mm-thick 3D-CT image slices preoperatively and at the 1-year postoperative follow-up; 3D-CT scans were evaluated using the same methodology as in our previous study [19]. To measure

A.　　　　　　　B.　　　　　　　C.

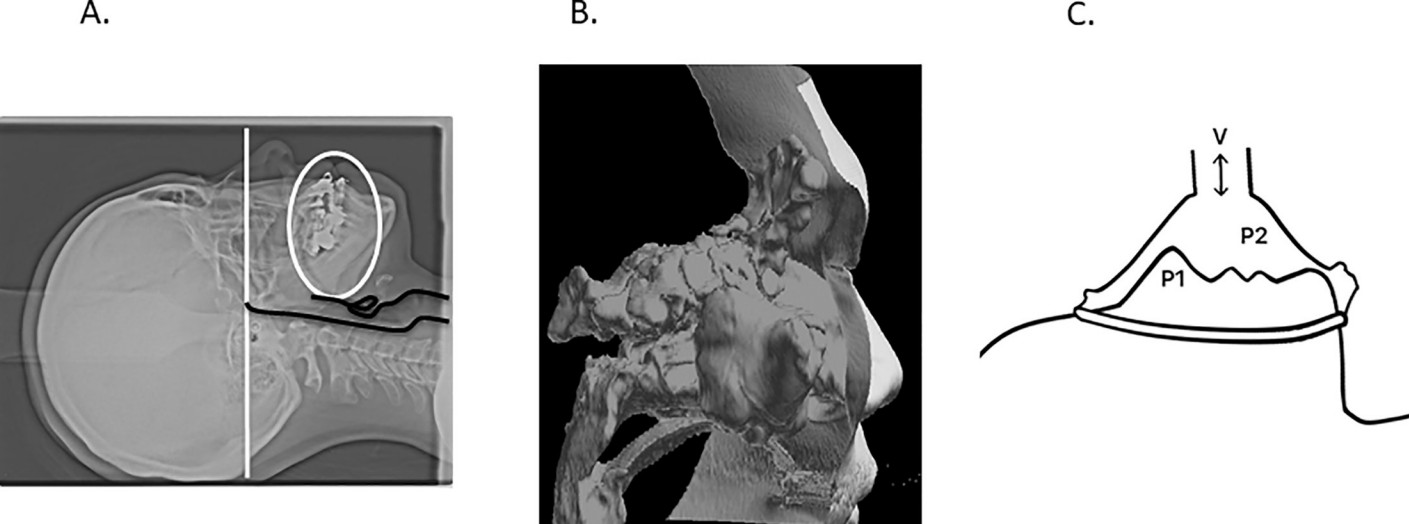

**Fig 1. Airway geometry and rhinomanometric measurement methods.** A: To measure airway geometries of all cases in the same condition, X-ray computed tomography (X-CT) scanning measurements were conducted using the Frankfort horizontal plane (white line) so that it was perpendicular to the floor in a maximal intercuspal position, with mouth closed, the tongue pressing against the palate (white circle); B: The extracted airway geometry data were transformed into text files of Stereolithography (STL) format; C: Rhinomanometric measurements (i.e., nasal patency inspection) were also conducted before and after surgery operations in accordance with the anterior-mask method.

airway geometries of all cases in the following condition, X-ray CT (X-CT) scanning measurements were conducted. One of the eight patients' data was shown in Fig 1. The participants laid supine using the Frankfort horizontal plane (white line) in a maximal intercuspal position, with mouth closed, the tongue pressing against the palate (white circle), and breathing quietly without swallowing (Fig 1A). Airway geometry data, including those for the accessory nasal sinuses and face surfaces, were extracted from X-CT DICOM data using the 3D volume rendering software Intage Volume Editor (version 1.1; Cybernet Systems Co., Ltd., Tokyo, Japan). The extracted airway geometry data were transformed into Stereolithography (STL) text files, as shown in Fig 1B. Rhinomanometric measurements (i.e., nasal patency inspection) were also conducted before and after surgery operations in accordance with the anterior-mask method, as shown in Fig 1C. Regarding rhinomanometry, the anterior-mask method was used to evaluate separate measures of the nasal resistance for the right and left nares. Using a formula based on Ohm's law of parallel resistance, the total nasal resistance was calculated using the measurement data of both the right and left nasal airways. Data on F, P1, and P2 were needed to obtain the right naris' R. If the left nasal cavity (i.e., on the opposite side) was not completely blocked, P2 could be derived from the left holorhinal route. Therefore, regardless of which naris is blocked, the total nasal resistance can be calculated using the mathematical formula for computing the equivalent resistance based on Ohm's law. Three-dimensional reconstructed STL models included the facial surface, nasal cavity, paranasal sinuses, and upper airway (until the tip of the epiglottis) but excluded the soft tissue surrounding the upper airway. The surfaces were corrugated because of artifacts caused by digitization and were, therefore, smoothed to facilitate computational meshing [20].

Fig 2A and 2B show one of the eight patients' data, which was unstructured CFD meshes generated for this study. Fig 2A shows the surface mesh structure from a side view, and Fig 2B shows the 3D-structure of all mesh cells from a transparency view. All CFD meshes for the current participants were prepared from the STL geometry data with the mesh generation software HEXPRESS (version 7.2; NUMECA International Company, Brussels, Belgium). To

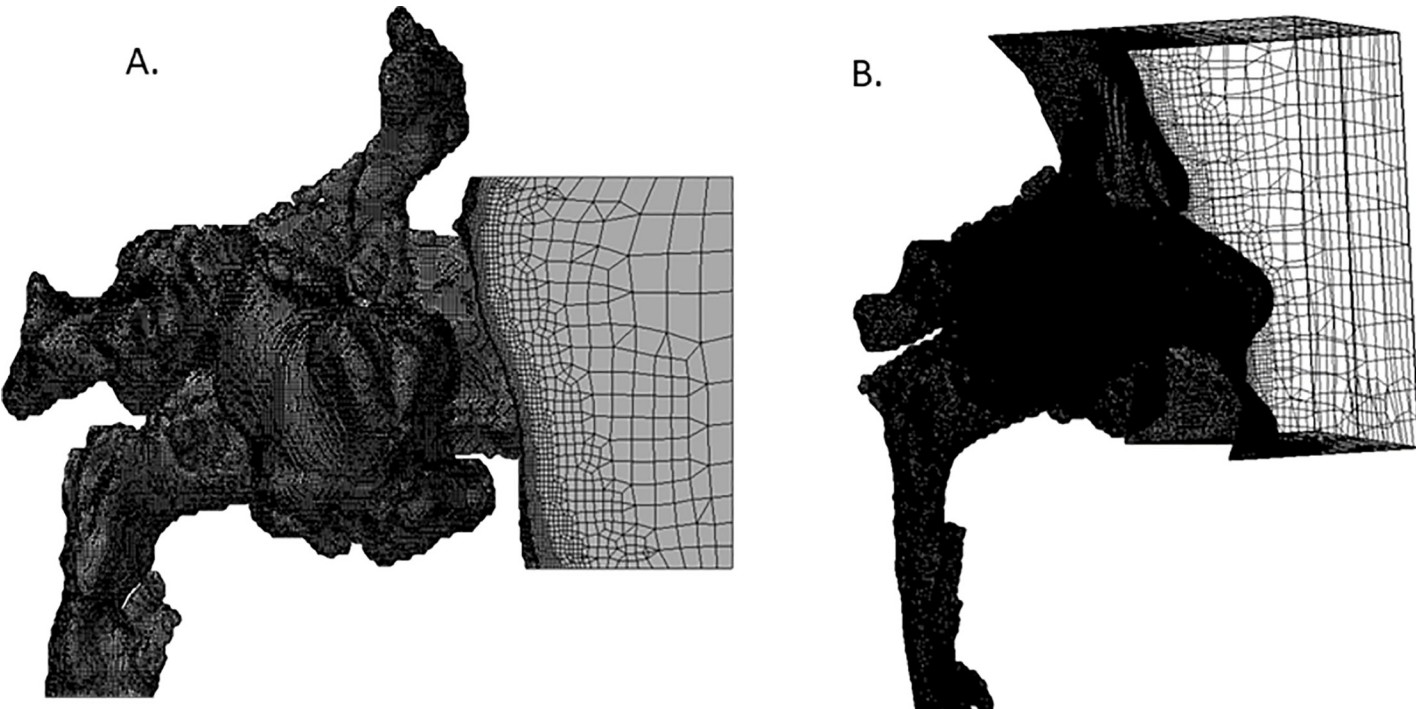

**Fig 2. Computational Fluid Dynamics (CFD) mesh structure.** A: The surface mesh structure from a side view; B: The three-dimensional structure of all mesh cells from a transparency view.

reproduce inlet airflows accurately, coming from outside the external naris and into the nasal airway, the calculation mesh was accurately formed from the space in front of the face to the nasal cavities and pharynx airway. Since our previous CFD studies showed that small, but non-negligible, swirling flows existed in the paranasal sinuses, the CFD meshes for this study also included the paranasal sinuses [16, 19]. Furthermore, the surrounding four plane surfaces of box-like spaces before the face surface were defined as slip solid surfaces because they were used only for limiting the CFD domain to prevent unnecessary calculations of very low-speed flow far from the nostrils [14]. For inspiratory flow CFD analysis, the inlet boundary was set on a front plane (right-hand side of Fig 2A) of the anterior region of the face, while the outlet boundary was set on a cross-section of the lower airway. For expiratory flow CFD analysis, both the inlet and outlet boundaries were interchanged. To analyze precisely the airway flows in the boundary layers of the nasal airway, upper pharyngeal wall, and facial surface, the total number of selected computational mesh points was large enough to produce a sufficiently fine hexahedral mesh close to the wall surface. The generated CFD meshes for all cases in this study comprised 3.6–15.0 million mesh cells. The y+ non-dimensional distance of the mesh points situated adjacent to the wall was kept between 2–5 for each CFD mesh.

For the CFD analysis, the CFD package software FINE/Open with Open Labs (version 7.2; NUMECA Co.) was used, and a steady-state solver was employed for Navier-Stokes equations with turbulence models. The Spalart-Allmaras one-equation turbulence model was used with the extended wall function. The maximum Reynolds number in this study was approximately 10000–25000. This study's analysis conditions of respiratory airflows were designed to model patient breathing at rest at atmospheric pressure and temperature of $1.013\times10^5$ Pa and 25°C, respectively. The coefficient of breathing air viscosity was $1.822\times10-5$ Pa·s. The non-slip and adiabatic conditions were used for the wall boundary on airway surfaces, except for the inlet

box described above. For the boundary conditions, the mass flow rate was fixed at the inlet, and static pressure was fixed at the outlet.

The convergence of the CFD calculations was determined based on the criteria that the average residual of CFD iterations should be less than $10^{-6}$, or the mass flow rate difference between inlet and outlet boundaries should be less than 0.5%.

In four out of the eight patients for whom meshes were prepared, the rhinomanometric measurements (i.e., nasal patency) were relatively favorable. For these four cases, CFD analysis was performed using the prepared meshes to simulate the nasal airway of rhinomanometric measurement conditions, either with the right or the left naris blocked for pressure measurements at the inlet and exit nasal airway at the same instance.

## Preoperative prediction of advantages of airway changes after maxillomandibular advancement surgery

Both a methodology and a prototype computational program were developed for this preoperative prediction study of airway change advantages after MMA surgery, and they were evaluated using the clinical data of this study. Fig 3 shows this process flow overview. The left side demonstrates the process flow of the present simulations, while the right side demonstrates typical processes of patients with jaw deformities who underwent MMA+GA surgery. The arrowed lines to the left denote data flow from our clinical practice to provide verification data, while the arrowed lines to the right denote the intended information of the study, as previously described.

The first process, as shown in Fig 3, is a CFD analysis using preoperative measurement data from clinical practice. The CFD methodology is described above; its accuracy has been proven in a previous paper [19]. Since MMA+GA surgery may enlarge the cross-sections near the exit of the nasal airway, nasopharynx, and oropharynx, both preoperative and postoperative rhinomanometric evaluations in the same patient can be used to assess improvements in nasal patency following surgery.

Both pre- and postoperative CFD analyses showed pressure and velocity distributions along the nasal airway, nasopharynx, and oropharynx; these results could provide useful data regarding the advantages of MMA+GA surgery.

The airway geometry deformation simulation process to predict surgical advantage was developed for this study. Geometric predictions were performed using preoperative STL data; three pivotal planes were introduced for the developed geometry deformation simulation program (Fig 4 shows an example group of three pivotal planes). These planes were defined using the right-handed Cartesian coordinate system, with the same unit and three-dimensional direction definition of the X-CT DICOM data used in our hospitals. The directions of x, y, and z are from right to left, from the face to the back, and from the foot to the head, respectively. The unit of length is in mm, and the three pivotal planes are orthogonal to the z-axis and are on the x-y plane. Above the top (z maximum) pivotal plane, the original geometry is fixed, and all points with larger y coordinates than the spine-side pharynx inner wall are fixed to the original positions. Enlargement of pharynx cross-sections, both on the middle and the lower pivotal planes, is controlled by four parameters (two dx and two dy parameters on the two planes): the increase of the anteroposterior airway diameter (dy) in the sagittal direction and increase of the airway width (dx) in the horizontal direction on the middle and the lower pivotal planes. The deformation parameters dx and dy on the two planes are configured with a preoperative prediction of airway change introduced from our practical accomplishments of MMA surgery. All points on the three pivotal planes and between these planes of the STL data, except for points of the fixed region, are smoothly displaced using combined three-

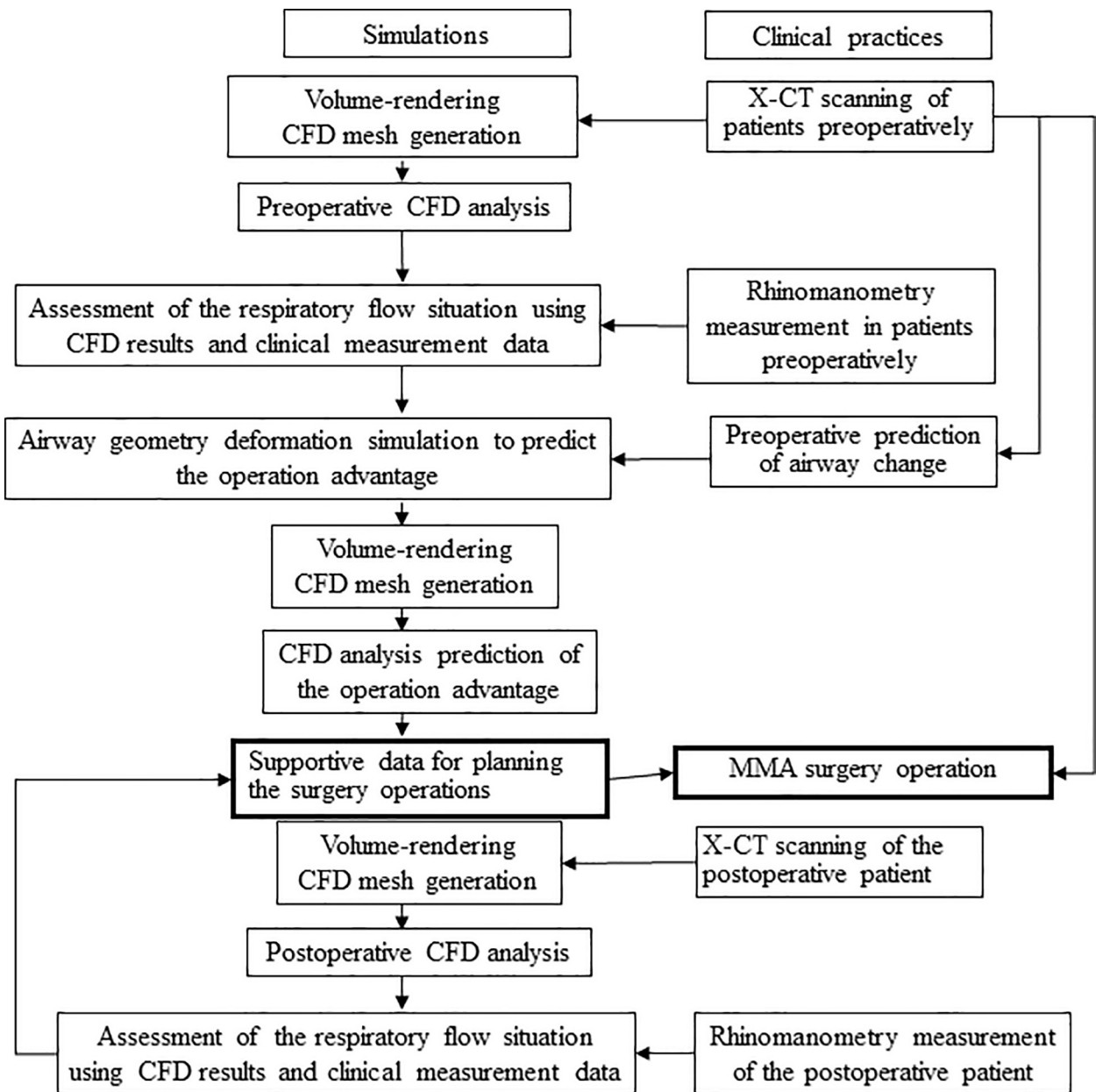

**Fig 3. Preoperative prediction of airway changes using the developed geometry deformation method, Computational Fluid Dynamics (CFD) analysis, and clinical measurement data.** Abbreviations: MMA, maxillomandibular advancement; X-CT, X-ray computed tomography.

dimensional quadratic equations. Since these combined equations are formulated to have continual differentials along all x, y, and z directions, the deformed airway wall is uniformly smooth. The locations of the three pivotal planes (i.e., sections 1, 2, and 3) are shown in Fig 4A. Section 1 is located on the horizontal plane through the left and right nasal confluence, section 2 is located on the horizontal plane through the tip of the soft palate, and section 3 is located on the horizontal plane through the tip of the epiglottis. Their airway cross-sections before surgery (surrounded by red circles) are shown in Fig 4B, 4C and 4D. The tip of the soft palate cross-section (Fig 4C) is considerably small compared with those of the other two sections.

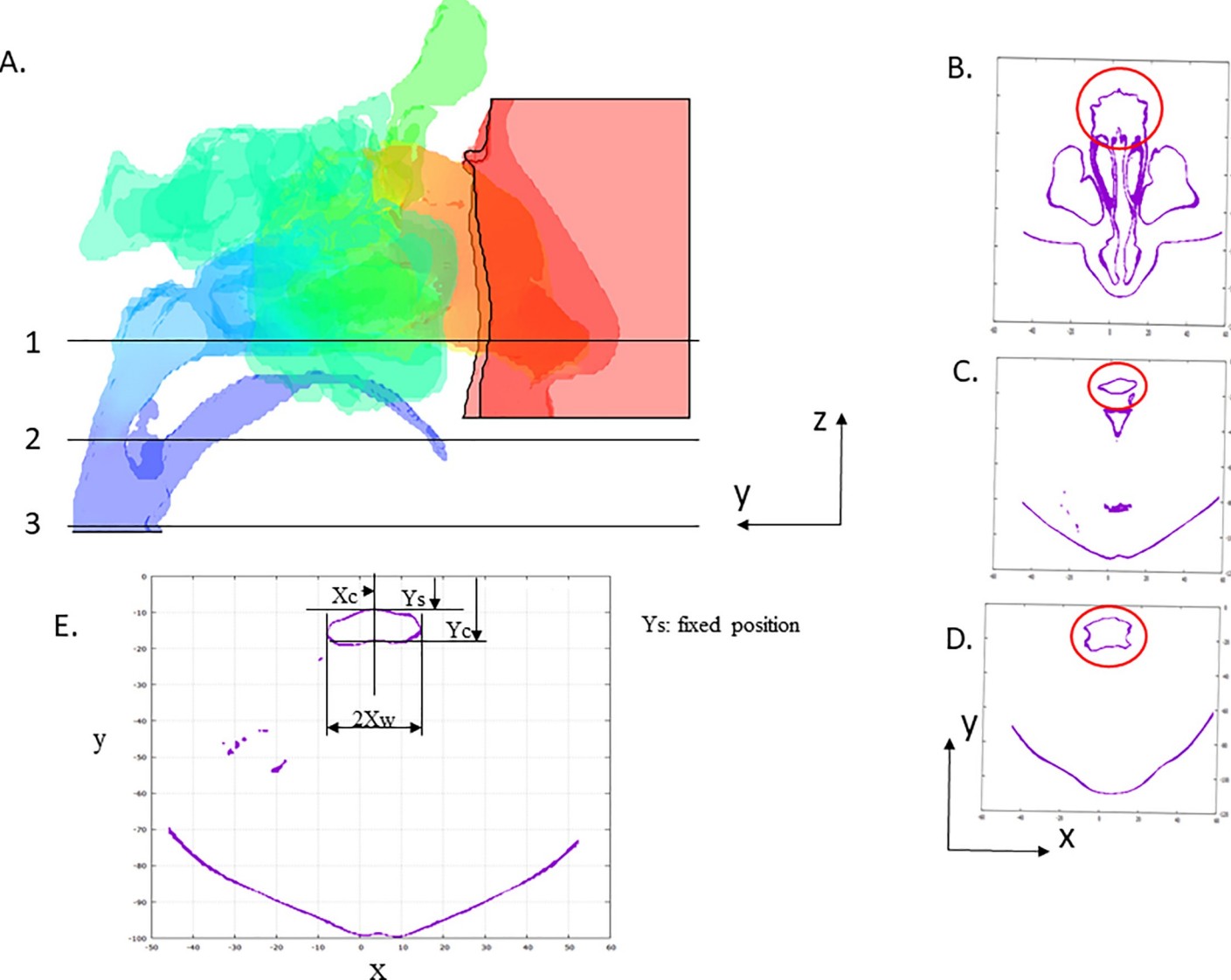

**Fig 4. Pivotal planes and definitions of the coordinates for airway geometry deformation simulations.** A: Pivotal planes for airway geometry deformation simulation. 1, Horizontal plane through the left and right nasal confluence. 2, Horizontal plane through the tip of the soft palate. 3, Horizontal plane through the tip of the epiglottis; B: The airway cross-sections before surgery on section 1; C: The airway cross-sections before surgery on section 2; D: The airway cross-sections before surgery on section 3. E: The digital 3D coordinates of the pharynx airway walls on arbitrarily selected horizontal planes for measurements and simulations.

## Method for prediction of pharynx deformations

The prototype computational program specifically developed for this study was designed to be used by oral surgeons. Consequently, the input data were able to be taken from surgical empirical data and were kept to the necessary minimum. The computational program was coded with FORTRAN 90 to allow usage on personal computers, workstations, and high-performance computation systems, including FUGAKU. Table 1 shows example input data of deformation parameters on the horizontal pivotal planes (Fig 4A) for a pharynx geometry deformation simulation. This program can provide digital 3D coordinate dimensioning of the pharynx airway walls on arbitrarily selected horizontal planes for measurements, drawing, and/or viewing for common graphing utilities, such as Gnuplot. Therefore, $Y_s$ (maximum y

**Table 1. Examples of input parameters for a pharynx geometry deformation simulation.**

| | Z | Ys | Yc | dY | dX (half to peak) |
|---|---|---|---|---|---|
| **Nasopharynx plane** | -610.00 | -7.78 | – | 6.00 | 2.00 |
| **Tip of the soft palate plane** | -587.00 | -11.76 | -20.00 | 4.00 | 1.00 |
| **Oropharynx plane** | -566.00 | All STL points are fixed both above and on this surface | | | |

Z, Z-axis; Ys, maximum y coordinate of the pharynx wall; Yc, minimum y coordinate of the pharynx wall on a sagittal plane; dY, the increase of an anteroposterior pharynx airway diameter in a sagittal direction on horizontal planes; dX, a half enlargement of a pharynx airway width in a coronal direction on horizontal planes from preoperative to postoperative geometry.

coordinate of the pharynx wall) and Yc (minimum y coordinate of the pharynx wall on a sagittal plane) can be calculated using this function of the program. "dX" denotes a half enlargement of a pharynx airway width in a coronal direction on horizontal planes from preoperative to postoperative geometry. "dY" denotes the increase of an anteroposterior pharynx airway diameter in a sagittal direction on horizontal planes as well. The dx and dy deformation (enlargement) parameters on the two planes can be selected from surgical knowledge.

Regarding the case shown in Table 1, the preoperatively predicted movements included forward movement of the upper jaw by 5 mm, upward movement of the axillary molar region by 2 mm, and forward movement of the chin by 6 mm. The expansion of soft tissues was estimated based on movements of the upper jaw, axillary molar region, and chin by the oral surgeon; the airway cross-section 2 would expand by 4 mm in the forward direction (Y-axis) and 2 mm in the lateral direction (X-axis), while cross-section 3 would expand by 6 mm in the forward direction (Y-axis) and 4 mm in the lateral direction (X-axis).

## Results

### Determining the optimal number of computational mesh points

STL data from a single representative patient were used to construct five meshes varying in size based on which the number of computational mesh points needed for analysis was determined. The numbers of computational mesh points and pressure differences between the inlet boundary (nasal airway inlet) and the outlet boundary (pharynx outlet) as main results of the CFD inspiratory airflow analysis ($\Delta$P and Pa, respectively) are shown in Fig 5. For each number of computational mesh points, Pa is shown on the Y-axis, and the total number of cells is shown on the X-axis. A mesh constructed from 2.0 million cells yielded 104 Pa and a y+ of 4.65602. At 3.5 million cells, the mesh-generated Pa and y+ were reduced to 39 Pa and 3.36864, respectively. At 7.0 million cells, the mesh produced 25 Pa and a y+ of 3.93312. At 10.3 million cells, the Pa and y+ were slightly reduced to 22 Pa and 3.88864, respectively. Finally, at 16.5 million cells, the Pa and Y+ were 20 Pa and 3.94373, respectively. Because computational meshes were generated to keep necessary density and suitable distances from the airway walls around boundary layers, the y+ non-dimensional distance of the mesh points situated adjacent to the wall was kept between 2–5 for each CFD mesh. This study for grid refinement suggests that the preferable total number of mesh cells is 10 million or more.

### Rhinomanometric measurements and simulation results

The primary aim for the anterior rhinomanometric method was to measure the nasal resistance separately in the right and left nares. Using a formula based on Ohm's law of parallel resistance, these measurements may be used to calculate total nasal resistance. The right and left nasal resistance values are, therefore, actual measurements, whereas the total nasal

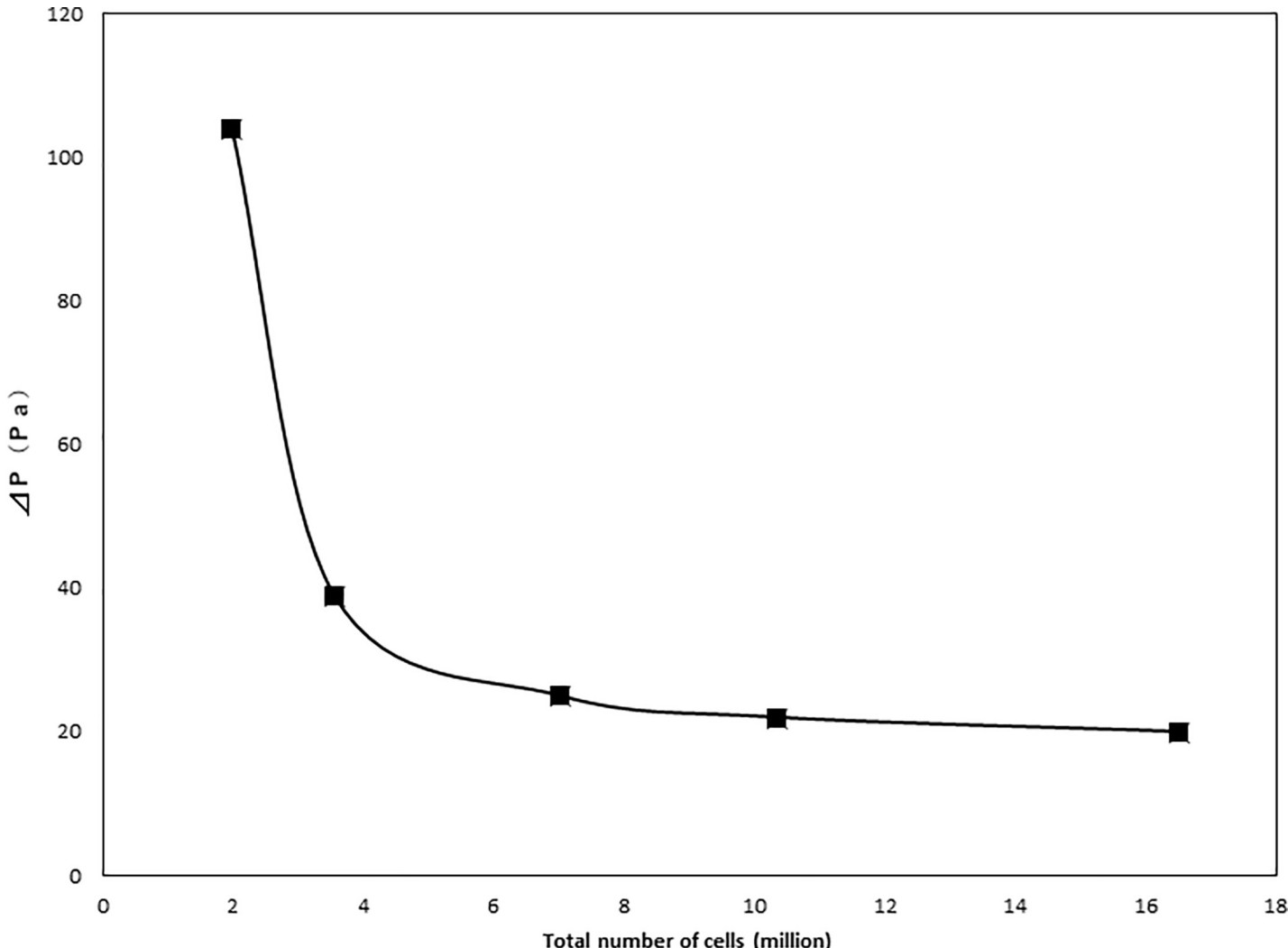

**Fig 5. Grid refinement study.** Vertical axis (⊿P) shows pressure differences between the inlet boundary (nasal airway inlet) and the outlet boundary (pharynx outlet); horizontal axis shows the total numbers of computational mesh points.

resistance is a calculated value. To obtain the right nasal resistance (R), measurements for the right nasal airflow (F), the right nasal anterior pressure (P1), and the posterior pressure (P2) are needed. Since the left naris (i.e., on the opposite side) was completely blocked with a plug through which a static pressure tube was inserted, it was possible to derive P2 (the measured static pressure at the outlet of the right nasal airway from this static pressure tube via the left holorhinal route).

Fig 6 shows comparisons between the rhinomanometric measurements and simulation results during exhalation and inhalation of the right and left nasal airways of the Case I patient. The measurement and CFD analysis were performed using the same conditions. During the measurements and CFD analyses for one side nasal airway, the other side nasal airway was blocked at the nostril inlet. The pressure difference between the inlet and outlet sections of the nasal airway is shown on the X-axis, and the volume flow ($cm^3/s$) is shown on the Y-axis. CFD analysis was performed with the inlet boundary condition as the mass flow rate and the outlet boundary condition as static pressure. Pre- and postoperative rhinomanometric results are

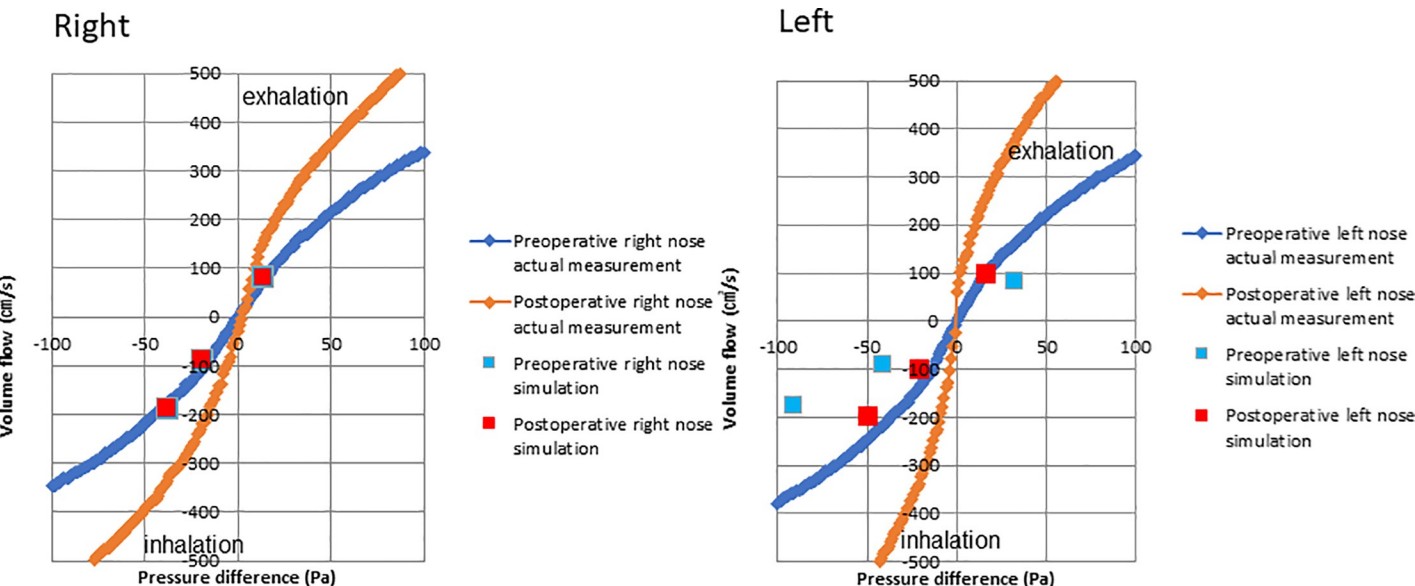

**Fig 6. Results of the computational simulation of fluid dynamics and rhinomanometry measurements of case I patient during inhalation and exhalation in the right and left nose.**

shown by the blue and red linked points, respectively. The results for the right and left nares are separately shown in Fig 6. The measurements showed that both nares' patency had been clearly improved, but the simulation results predicted no improvement for the right naris; this might be due to a blockage caused by a condition such as allergic rhinitis found in the right naris, as observed by CT scans. From the clinical findings of this Case I patient, it is possible that the condition of the right naris blockage for the duration of the X-CT was different from that of the rhinomanometry measurement duration. Considering the clinical diagnosis of the right naris, overall, both the preoperative and postoperative data were qualitatively consistent between the simulation results and postoperative measurements.

### Evaluating the static pressure distributions in the right and left nares

Fig 7 shows the inhalation CFD results of static pressure distributions (unit Pa) on the sagittal planes before and after surgery in the right nasal airway (A) with the left nostril plugged and in the left nasal airway (B) with the right nostril plugged. The volume flow rate was around 200 cm³/s for all analyses.

From the bottom view, the static pressure on the side with the blocked nasal cavity was approximately the same as the static pressure at the outlet boundary. For both the right and left nares, Pa was compared between the blocked inlet boundary and the inside of the nasal cavity on the blocked side. In the Case I patient, the left nostril's pressure was approximately 94 and 59 Pa before and after surgery, respectively. Moreover, for the right nostril, the pressure was approximately 36 and 62 Pa before and after surgery, respectively. As shown in Fig 7, the left nasal airway geometry after the MMA surgery remained similar to that before the surgery. On the other hand, the pharynx airway, including the nasal pharynx, was enlarged. In addition, the narial inlet and paranasal sinuses' static pressures were clearly reduced after surgery. As discussed previously, the postoperative right upper and middle nasal meatuses were blocked (Fig 7). Since this blockage was a transient symptom, it was not associated with the surgery. Thus, in the Case I patient, the surgery was successful in reducing the left nasal pressure difference.

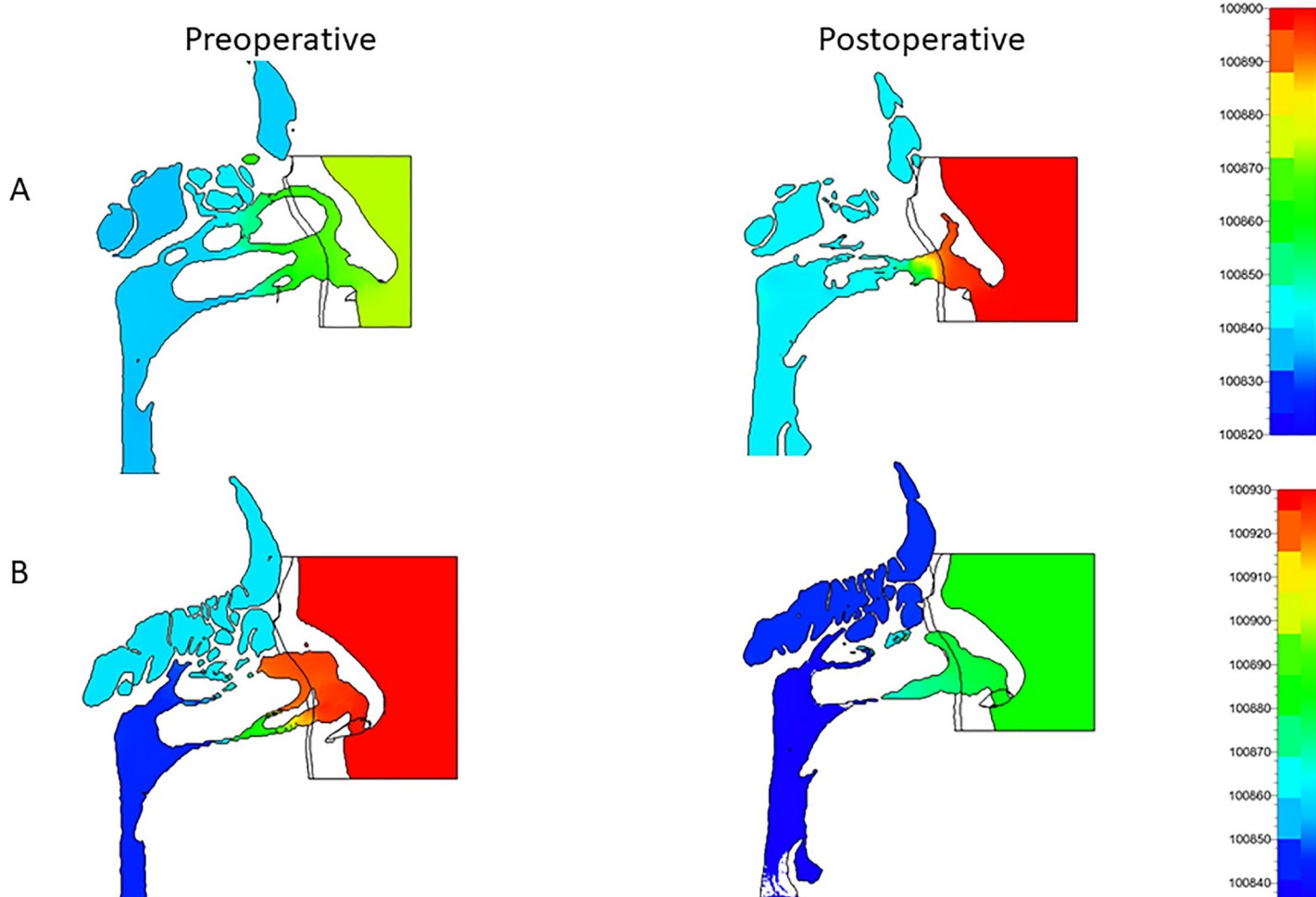

**Fig 7. Preoperative and postoperative static pressure in the right (A) and left (B) nostrils for the Case I patient on the sagittal plane.**

## Simulated pressure and streamline distributions

Figs 8 and 9 show the static pressure distributions and streamlines with the magnitude of velocity in the right and left nasal cavities and the pharynx away from the bottom view for the Case I patient. Regarding the Case I patient, in the right nostril, the preoperative and postoperative maximum velocity was 2.8 m/s and 5.5 m/s, respectively; in the left nostril, the preoperative and postoperative maximum velocity was 4.2 m/s and 3.5 m/s, respectively. In general, postoperative maximum velocities increased following surgery; however, when the flow velocities in the airway regions were compared, airflow was straightened, and the overall flow velocity was decreased. This finding indicates that the surgery reduced airflow pressure loss. From the bottom view, the static pressure on the side with the blocked nasal cavity was approximately the same as the static pressure at the outlet boundary (Figs 8A and 9A).

Our findings indicate that the nasal and pharynx airways have some sudden expansion flow paths, where the static pressure decreases suddenly and the flow velocity is accelerated, thus inducing jet flows and flow separations (Figs 7–9). Since these phenomena cause pressure losses, it is possible that the calculated pressure difference error can be reduced by increasing the total number of CFD mesh cells while the y+ is not greatly changed (Fig 5).

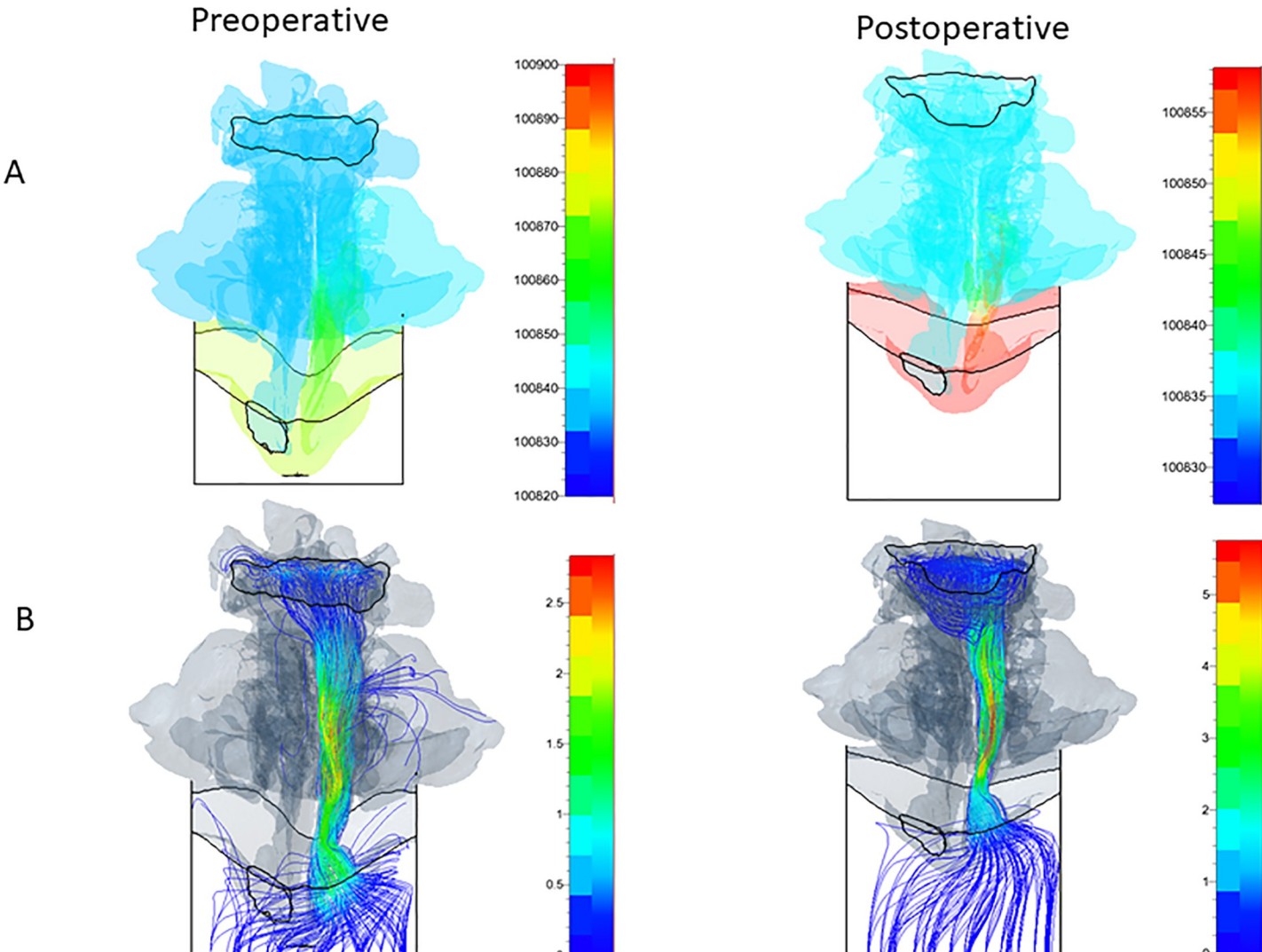

**Fig 8.** Preoperative and postoperative static pressure (A) and streamline with the magnitude of velocity (B) in the right nose for Case I patient from the bottom view.

### Visualizing the static pressure distributions

With the right naris blocked and air flowing only through the left naris, static pressure distributions were visualized at cross-sections 1 cm from the nasal cavity inlet and 1 cm above the outlet (S1 and S2 Figs). The outlet pressure was fixed at 100,836 Pa and, therefore, approximated the outlet pressure before and after surgery. On the right nostril, comparing the cross-sections obtained 1 cm from the nasal cavity inlet in the Case I patient, the static pressure decreased from 100,972 Pa before surgery to 100,877 Pa after surgery (S1 Fig). On the left nostril, in the case of the cross-section 1 cm from the nasal cavity inlet, the static pressure increased from 100,871 Pa before surgery to 100,894 Pa after surgery (S2 Fig).

### Simulating postoperative airway changes and CFD analysis

Notably, this study is the first to perform the airway deformation simulation and CFD analysis using the deformed geometry discussed below. The surgery of the patient whose clinical data were used was completed before this simulation study.

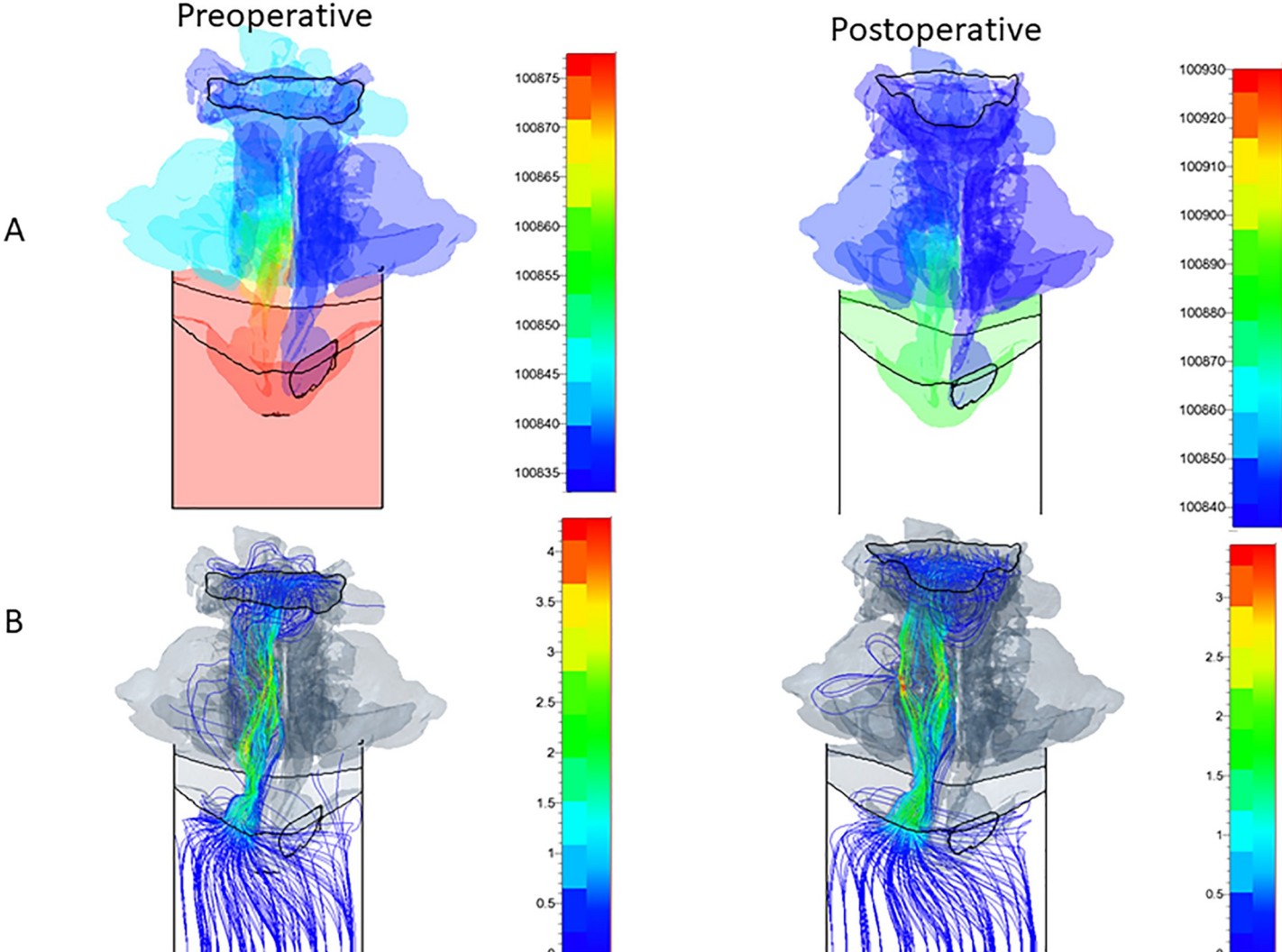

**Fig 9.** Preoperative and postoperative static pressure (A) and streamline with the magnitude of velocity (B) in the left nose for Case I patient from the bottom view.

A horizontal cross-section of the geometry prediction model is shown in Fig 4. Pre- and postoperative cross-sections measured with X-CT on the same x-y horizontal plane are shown in Fig 10A and 10C, respectively. The lower direction is the face side (-y direction), where all three cross-sections are depicted on the same scale. The increase of the anteroposterior airway diameter (dy) in the sagittal direction from preoperative (Fig 10A) to the predicted (Fig 10B) values was 3.6 mm, while the dy input deformation parameter was 4.0 mm. Since the developed deformation algorithm was designed to maintain three-dimensional smoothness while allowing small changes in the dy input, the dy of the simulated deformed pharynx was not strictly identical as in the input data. Conversely, the increase of the anteroposterior airway diameter (dy) in the sagittal direction from preoperative (Fig 10A) to postoperative (Fig 10C) was 6.8 mm. To evaluate the ability to generate enlarged three-dimensional airways with target widths and/or target areas at any horizontal plane, the deformation program was used to generate a similar airway as the postoperative three-dimensional geometry of the pharynx. Fig 10D shows the cross-section of the oropharynx at follow-up prediction, and Fig 10E shows the real postoperative cross-sectional geometry of the oropharynx for comparison purposes. In

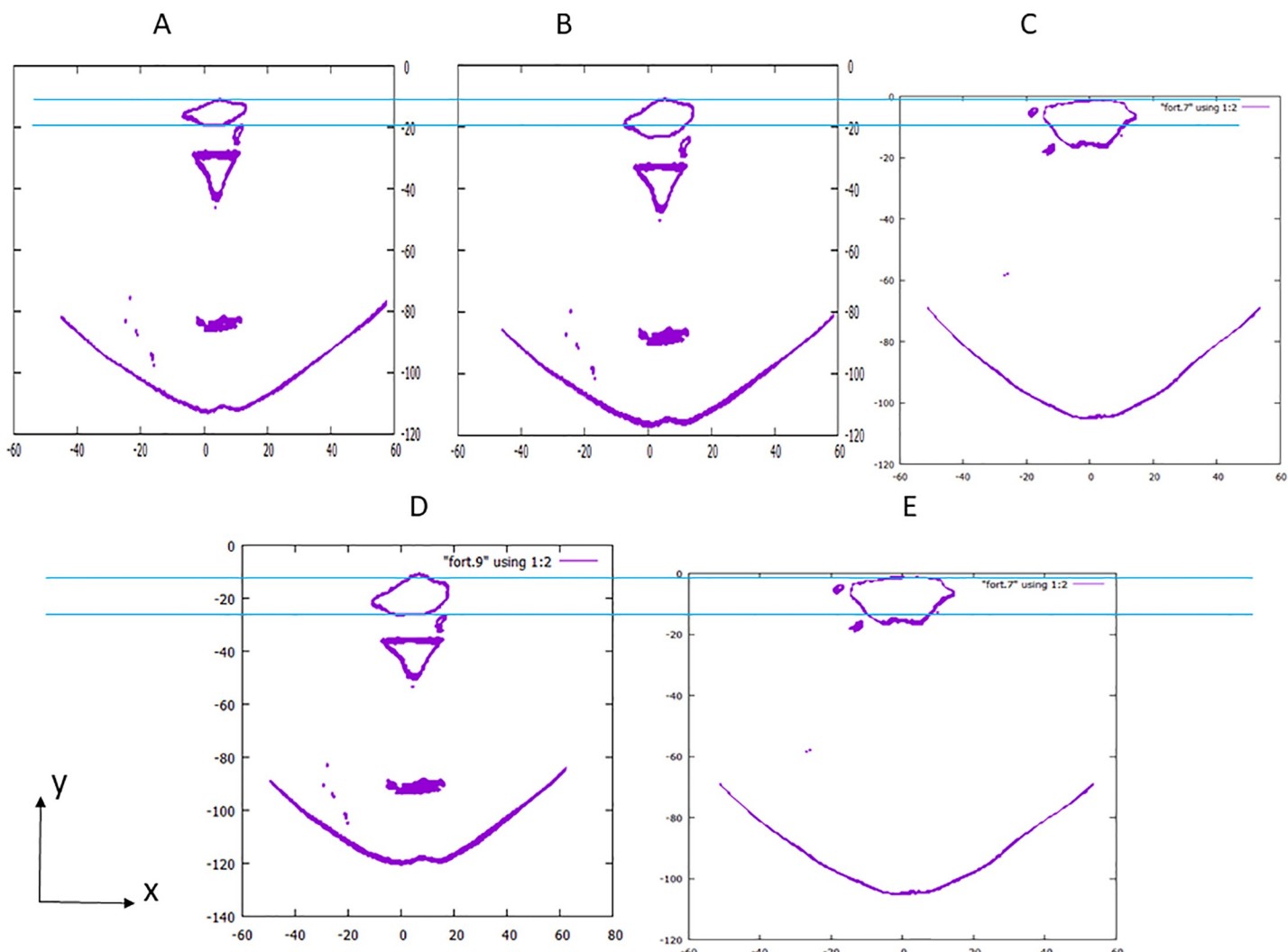

**Fig 10. Comparison between the preoperative, prediction, and postoperative geometries on the horizontal plane of the tip of the soft palate.** A: Preoperative geometry measured with X-ray computed tomography (X-CT); B: Prediction geometry deformed with algebraic equations; C: Postoperative geometry measured with X-CT; D: Follow-up prediction geometry deformed with algebraic equations; E: Postoperative geometry measured with X-CT.

addition, Fig 10E shows the same geometry as Fig 10C. The y- and x-direction widths of the predicted cross-section became the same as those of the postoperative geometry after a couple of quick computation iterations. The follow-up predicted geometry with the same area as the postoperative geometry was similarly obtained from the original STL data. However, Fig 10D and 10E show that, in particular, the shape of the postoperative and the follow-up predicted geometry differed around the spine-side airway wall because of the limitation of the algorithm in the development program.

Fig 11 shows the pressure difference between preoperative, predicted, and postoperative CFD results, which was obtained by comparing the rhinomanometric measurement data of the same patient shown in Fig 10A–10E, i.e., Case II patient. The CFD results of both inhalation and exhalation conditions showed that the pressure differences decreased after MMA surgery. Moreover, the CFD-analyzed pressure differences of the predicted airway decreased from the pressure differences of the preoperative results. The pressure differences of the

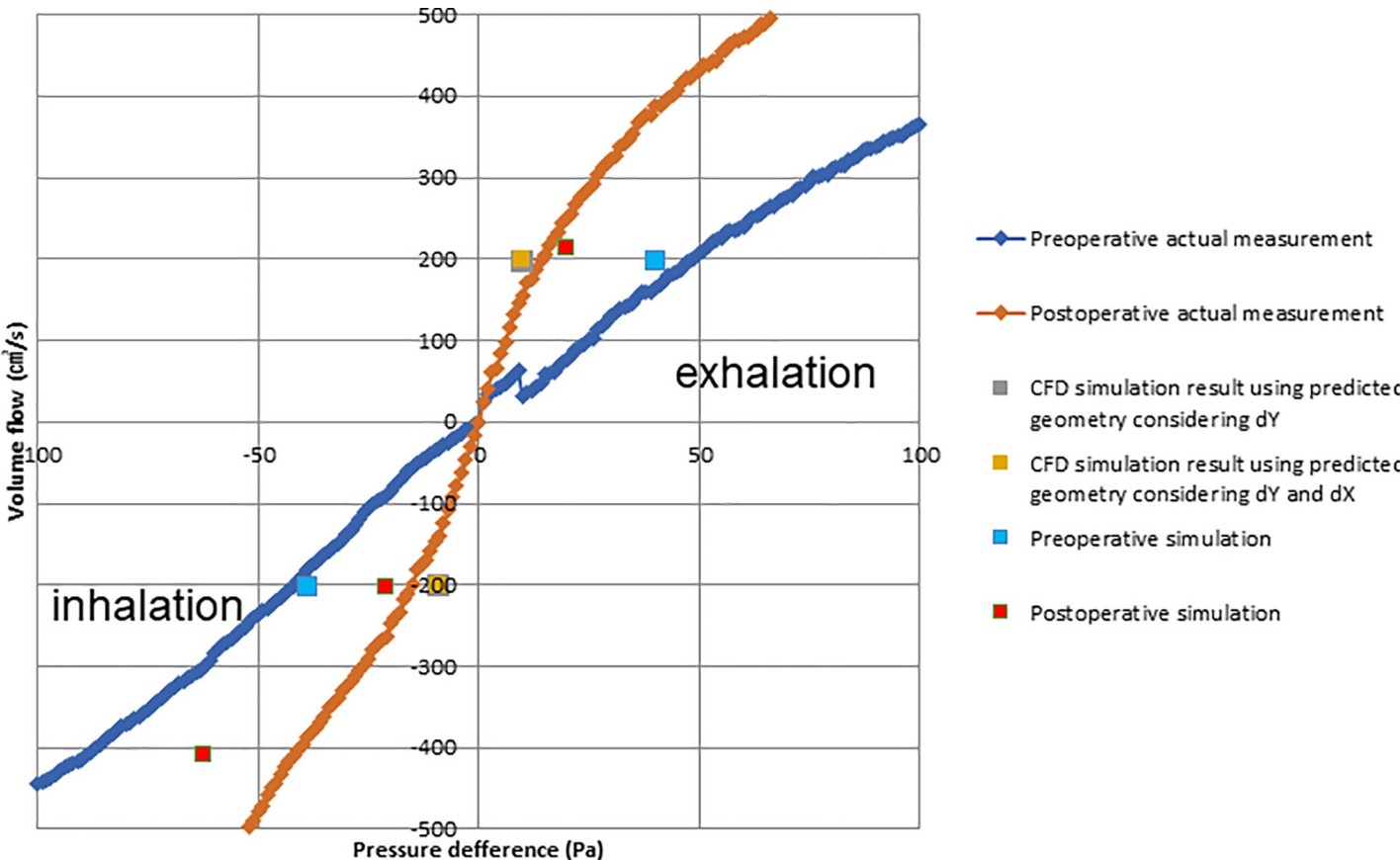

**Fig 11. Pressure difference Computational Fluid Dynamics (CFD) results of the preoperative, predicted, and postoperative values, as assessed by comparing the rhinomanometric measurement data.** Case II patient.

predicted airway were lower than those of the postoperative results in both the inhalation and exhalation conditions.

Fig 12A and 12B show the velocity distributions on the oropharynx cross-sections in the same pre- and postoperative CFD analyses as shown in Fig 11. The computed velocity distributions are visualized with color contours (right) and two graphs (left) along the x- and y-direction lines on the cross-section. Both the peak and averaged velocities were clearly reduced by the MMA surgery. Since the velocity distributions mainly show local aerodynamic phenomena, the difference of the velocity distributions between the preoperative and postoperative analysis results is visible; this decrease in velocity should induce negative pressure in the oropharynx airway. Fig 12C shows the velocity distributions on the oropharynx cross-section of the same predicted airway CFD analysis as shown in Fig 11. Since the predicted section area was smaller than the postoperative area, the reduction in velocity was moderate when compared to the postoperative result.

## Discussion

In this study, a geometry prediction simulation methodology was developed based on our previous work. This methodology provided artificial STL data for prediction using CFD analyses, and our clinical findings of nasal and pharyngeal movements after MMA surgery were further used for comparison analysis.

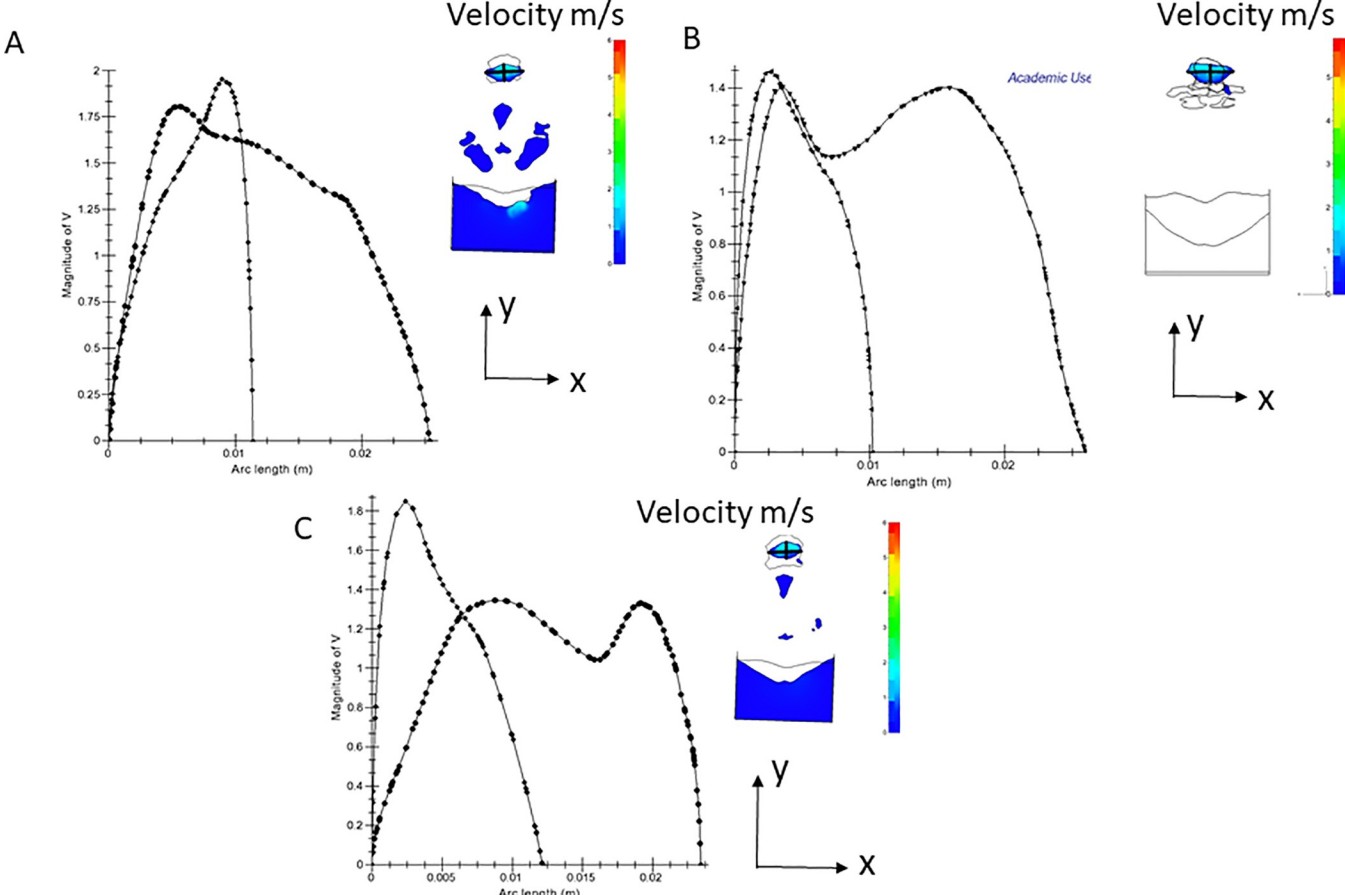

**Fig 12. Velocity distributions in the oropharynx cross-section of the preoperative Computational Fluid Dynamics (CFD) analysis in the case II patient.**
A: Horizontal plane through the left and right nasal confluence; B: Horizontal plane through the tip of the soft palate; C: Horizontal plane through the tip of the epiglottis.

To identify airway structures that may lead to OSA, previous studies have used both cephalometry and CT. Recently, CFD has become a highly precise and reliable method for numeric analysis of respiration flow and has emerged as the preferred tool for predicting OSA [14, 16, 21, 22]. However, a CFD analysis requires large-scale calculations [15], and there are precision-related difficulties with comparing simulation results to measured physiological data. Notably, in this study, the actual rhinomanometric measurements were compared with simulation results.

Our previous studies [19, 20, 23] demonstrated that CFD simulations could largely reproduce patients' actual respiratory flow dynamics. This study suggests that CFD can be used to analyze respiration air flows in areas that cannot be measured by rhinomanometric methods, i.e. the pharynx airway, and after the preoperative confirmation of both airway stricture and high static pressure sites, warnings should be provided regarding potential airway blockage. In addition, it has also been suggested that movement during surgery should be taken into consideration [19].

## Computational mesh points

The first step of CFD analysis is to determine the appropriate number of computational mesh points and compare CFD results using meshes of different total numbers of mesh cells (Fig 5).

As shown in Fig 5, the pressure difference increased with a decreased number of computational mesh points; however, it did not markedly differ at higher numbers of computational mesh points (i.e., >7-million cells). Nevertheless, regarding the mesh with 3.5-million cells, the computational analysis needed approximately 4 hours to complete; this was significantly less than the 12 hours needed for analyzing the mesh with 16.5 million cells. Therefore, considering both the computational time and analysis efficiency, the CFD meshes with 10 million cells or more were selected for the current study.

### Rhinomanometry measurement, static pressure, and airflow velocity

This study found that the levels of improvements in airway conditions and nasal patency of patients who received orthognathic surgery were consistent with those reported in a previous study [19]; however, that study analyzed only inhalation. Conversely, our study analyzed both inhalation and exhalation, thus improving the accuracy by which patients' respiratory states could be measured. Importantly, simulation results and actual measurements were approximately consistent for both inhalation and exhalation. Here, simulations were performed with either the right or left naris blocked, and the predictions were similar to those found clinically. As shown in Fig 7, the trend in the postoperative measurements demonstrated an increase, but the patient showed nasal fluid accumulation during CT, suggesting the possibility of nasal blockage. As shown in Figs 8 and 9, the postoperative flow velocity was more homogeneous than the preoperative flow velocity. These results indicate that surgery reduces airflow pressure loss.

### Deformation simulation

In the geometry prediction simulations using preoperative STL data for the airway alone, preoperatively predicted movement was used to extrapolate postoperative airway expansion (estimated visually). Using this visually estimated value, a geometry prediction model was generated from the preoperative STL data. Geometry prediction simulation results were compared with the real postoperative STL data and postoperative simulation results. Even for geometry prediction simulations, an analysis could be performed without airflow stagnation between the inlet and outlet. Static pressure was qualitatively consistent between the geometry prediction and the postoperative simulations. Furthermore, both the geometry predictions and postoperative simulation results were consistent with postoperative measurements. These findings suggest that the geometry prediction simulation methodology presented in this study is useful for preoperatively predicting the postoperative structure and providing supportive data for planning surgery operations.

### Supporting information

**S1 Fig.** Preoperative and postoperative static pressure on the outlet (A) and inlet (B) in the right nasal passage for the Case I patient.
(TIF)

**S2 Fig.** Preoperative and postoperative static pressure on the outlet (A) and inlet (B) in the left nasal passage for the Case I patient.
(TIF)

### Acknowledgments

We wish to thank Editage (www.editage.jp) for English language editing.

## Author Contributions

**Conceptualization:** Kanako Yamagata, Tadashi Tanuma, Morio Tonogi.

**Formal analysis:** Kanako Yamagata, Shouhei Ogisawa, Tadashi Tanuma.

**Funding acquisition:** Keiji Shinozuka, Tadashi Tanuma, Morio Tonogi.

**Investigation:** Kanako Yamagata, Akio Himejima, Hiroaki Azaki, Shuichi Nishikubo, Takako Sato, Tadashi Tanuma.

**Methodology:** Kanako Yamagata, Masaaki Suzuki, Tadashi Tanuma.

**Project administration:** Keiji Shinozuka, Morio Tonogi.

**Software:** Kanako Yamagata, Tadashi Tanuma.

**Supervision:** Morio Tonogi.

**Writing – original draft:** Kanako Yamagata, Keiji Shinozuka, Tadashi Tanuma.

**Writing – review & editing:** Tadashi Tanuma, Morio Tonogi.

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
