## [Decision Letter · Decision Letter 0]

16 Jul 2021

PONE-D-21-18930

A preoperative predictive study of advantages of airway changes after maxillomandibular advancement surgery using computational fluid dynamics analysis

PLOS ONE

Dear Dr. Shinozuka,

Thank you for submitting your manuscript to PLOS ONE. After careful consideration, we feel that it has merit but does not fully meet PLOS ONE’s publication criteria as it currently stands. Therefore, we invite you to submit a revised version of the manuscript that addresses the points raised during the review process.

We look forward to receiving your revised manuscript.

Kind regards,

Giannicola Iannella, M.D

Academic Editor

PLOS ONE

Journal Requirements:

3. You indicated that you had ethical approval for your study. In your Methods section, please ensure you have also stated whether you obtained consent from parents or guardians of the minors included in the study or whether the research ethics committee or IRB specifically waived the need for their consent.

4. Thank you for stating the following in the Acknowledgments Section of your "We wish to thank Editage (www.editage.jp) for English language editing. This study was supported by the Japan Society for the Promotion of Science (KAKENHI, no. 19K10294) grants from the Dental Research Center (2019, 2020, and 2021) and Sato Fund (2018), Nihon University School of Dentistry (Tokyo, Japan)."

"This study was supported by the Japan Society for the Promotion of Science (KAKENHI, no. 19K10294) grants from the Dental Research Center (2019, 2020 and 2021) and Sato Fund (2018), Nihon University School of Dentistry (Tokyo, Japan)."

6. We note that Figure(s) 1 and 2 in your submission contain copyrighted images. All PLOS content is published under the Creative Commons Attribution License (CC BY 4.0), which means that the manuscript, images, and Supporting Information files will be freely available online, and any third party is permitted to access, download, copy, distribute, and use these materials in any way, even commercially, with proper attribution. For more information, see our copyright guidelines: http://journals.plos.org/plosone/s/licenses-and-copyright.

a. You may seek permission from the original copyright holder of Figure(s) 1 and 2 to publish the content specifically under the CC BY 4.0 license. 

Reviewers' comments:

Reviewer's Responses to Questions

**Comments to the Author**

1. Is the manuscript technically sound, and do the data support the conclusions?

Reviewer #1: Yes

Reviewer #2: Yes

2. Has the statistical analysis been performed appropriately and rigorously? 

Reviewer #1: Yes

Reviewer #2: Yes

3. Have the authors made all data underlying the findings in their manuscript fully available?

Reviewer #1: Yes

Reviewer #2: Yes

4. Is the manuscript presented in an intelligible fashion and written in standard English?

Reviewer #1: Yes

Reviewer #2: Yes

5. Review Comments to the Author

Reviewer #1: The manuscript is well written, however minor corrections are needed to improve the quality.

Introduction

line 44, It has been shown that obstructive sleep apneas are also correlated with different comoribidities compared to the classic cardiovascular ones. In fact, a correlation has recently been reported between OSAS and nasal disorders both sensory with altered smell and inflammation the alteration of different demonstrable biomarkers such as oxygen free radicals, cell free DNA or lipid peroxidation. please cite the paper DOI:10.1007/s00405-020-06316-w and DOI:10.3390/jcm10020277

Methods

the methods used correct and well structured.

The results are consistent with the methods set out.

Figures are really interesting, representing well what the results presuppose.

Discussion correct and inherent.

Reviewer #2: Interesting paper, only few suggestions to improve the structure and make it more fascinable:

Introduction, after line 49:

- In the case of a retrolingual obstruction it is not always necessary to intervene through mandibular advancement, especially if there is a lymphatic lingual base hypertrophy or a primary epiglottis that can be modeled through a supraglottoplasty. please cite DOI:10.1002/rcs.2106

- A comparative stress test demonstrated excellent distribution of palatal vector forces after barbed surgery, reallocating and shaping the soft tissues in order to avoid collapsing of the walls. please cite DOI:10.1007/s00405-020-05883-2

Methods and Results really fascinating as the figures.

Good job!

6. PLOS authors have the option to publish the peer review history of their article (what does this mean?). If published, this will include your full peer review and any attached files.

Reviewer #1: No

Reviewer #2: No

---

## [Author Response · Author response to Decision Letter 0]

23 Jul 2021

Journal Requirements

Response:

We have confirmed it.

2. Please review your reference list to ensure that it is complete and correct.

Response:

We have double checked that the reference had been complete and correct.

3. You indicated that you had ethical approval for your study. In your Methods section, please ensure you have also stated whether you obtained consent from parents or guardians of the minors included in the study or whether the research ethics committee or IRB specifically waived the need for their consent.

Response:

We have added the sentence ‘Informed consent was obtained from parents/guardians of subjects < 20 years of ages.’ in the Materials and methods section (page 4, lines 86).

4. Thank you for stating the following in the Acknowledgments Section of your "We wish to thank Editage (www.editage.jp) for English language editing. This study was supported by the Japan Society for the Promotion of Science (KAKENHI, no. 19K10294) grants from the Dental Research Center (2019, 2020, and 2021) and Sato Fund (2018), Nihon University School of Dentistry (Tokyo, Japan)."

"This study was supported by the Japan Society for the Promotion of Science (KAKENHI, no. 19K10294) grants from the Dental Research Center (2019, 2020 and 2021) and Sato Fund (2018), Nihon University School of Dentistry (Tokyo, Japan)."

Response:

We have deleted the sentence "We wish to thank Editage (www.editage.jp) for English language editing. This study was supported by the Japan Society for the Promotion of Science (KAKENHI, no. 19K10294) grants from the Dental Research Center (2019, 2020, and 2021) and Sato Fund (2018), Nihon University School of Dentistry (Tokyo, Japan)." in the Acknowledgments Section.

Response:

I have linked my ORCID id to Editorial Manager account.

6. We note that Figure(s) 1 and 2 in your submission contain copyrighted images. All PLOS content is published under the Creative Commons Attribution License (CC BY 4.0), which means that the manuscript, images, and Supporting Information files will be freely available online, and any third party is permitted to access, download, copy, distribute, and use these materials in any way, even commercially, with proper attribution. For more information, see our copyright guidelines: http://journals.plos.org/plosone/s/licenses-and-copyright.

a. You may seek permission from the original copyright holder of Figure(s) 1 and 2 to publish the content specifically under the CC BY 4.0 license. 

Response:

We truly apologize confusing you because of our inappropriate expression. Figures 1 and 2 are our original data of our eight patients. That means these are not copyrighted images. As we re-wrote correctly, please check it again.

Reviewer’s comments

Reviewer: 1

Comments to the Author:

The manuscript is well written, however minor corrections are needed to improve the quality.

Comments: 

1. Introduction

line 44, It has been shown that obstructive sleep apneas are also correlated with different comoribidities compared to the classic cardiovascular ones. In fact, a correlation has recently been reported between OSAS and nasal disorders both sensory with altered smell and inflammation the alteration of different demonstrable biomarkers such as oxygen free radicals, cell free DNA or lipid peroxidation. please cite the paper DOI:10.1007/s00405-020-06316-w and DOI:10.3390/jcm10020277

Response:

We would like to thank the reviewer 1 who has reviewed our manuscript favorably and we appreciate the comments and suggestions to improve the content. As it has been suggested by the reviewer, we have made correction in the part of Introduction of manuscript. 

2. Methods

the methods used correct and well structured.

The results are consistent with the methods set out.

Figures are really interesting, representing well what the results presuppose.

Discussion correct and inherent.

Response:

We appreciate the comments.

Reviewer: 2

Comments to the Author:

Interesting paper, only few suggestions to improve the structure and make it more fascinable:

Comments:

1. Introduction, after line 49:

In the case of a retrolingual obstruction it is not always necessary to intervene through mandibular advancement, especially if there is a lymphatic lingual base hypertrophy or a primary epiglottis that can be modeled through a supraglottoplasty. please cite DOI:10.1002/rcs.2106

A comparative stress test demonstrated excellent distribution of palatal vector forces after barbed surgery, reallocating and shaping the soft tissues in order to avoid collapsing of the walls. please cite DOI:10.1007/s00405-020-05883-2

Response:

We would like to thank the reviewer 2 who has reviewed our manuscript favorably and we appreciate the comments and suggestions to improve the content. As it has been suggested by the reviewer, we have made correction in the part of Introduction of manuscript..

2. Methods and Results really fascinating as the figures.

Good job!

Response:

We appreciate the comments.

---

## [Editor Report · Decision Letter 1]

28 Jul 2021

A preoperative predictive study of advantages of airway changes after maxillomandibular advancement surgery using computational fluid dynamics analysis

PONE-D-21-18930R1

Dear Dr. Shinozuka,

We’re pleased to inform you that your manuscript has been judged scientifically suitable for publication and will be formally accepted for publication once it meets all outstanding technical requirements.

Kind regards,

Academic Editor

PLOS ONE

Additional Editor Comments (optional):

This is a very interesting study about airway changes after maxillomandibular advancement surgery using computational fluid dynamics analysis in OSA patients. The stusy is innovative and well conducted. Authors well improved the manuscript according to the comments of reviewers.

In my opinion it is suitable for the pubblication on PLOS-ONE.
---

## [Editor Report · Acceptance letter]

2 Aug 2021

PONE-D-21-18930R1 

A preoperative predictive study of advantages of airway changes after maxillomandibular advancement surgery using computational fluid dynamics analysis 

Dear Dr. Shinozuka:

I'm pleased to inform you that your manuscript has been deemed suitable for publication in PLOS ONE. Congratulations! Your manuscript is now with our production department. 

Kind regards, 

on behalf of

Dr. Giannicola Iannella 

Academic Editor

PLOS ONE